# BEYOND THE PIXELS: EXPLORING THE EFFECTS OF BIT-LEVEL NETWORK AND FILE CORRUPTIONS ON VIDEO MODEL ROBUSTNESS

## ABSTRACT

We investigate the robustness of video machine learning models to bit-level network and file corruptions, which can arise from network transmission failures or hardware errors, and explore defenses against such corruptions. We simulate network and file corruptions at multiple corruption levels, and find that bit-level corruptions can cause substantial performance drops on common action recognition and multi-object tracking tasks. We explore two types of defenses against bit-level corruptions: corruption-agnostic and corruption-aware defenses. We find that corruption-agnostic defenses such as adversarial training have limited effectiveness, performing up to 11.3 accuracy points worse than a no-defense baseline. In response, we propose *Bit-corruption Augmented Training* (BAT), a corruption-aware baseline that exploits knowledge of bit-level corruptions to enforce model invariance to such corruptions. BAT outperforms corruption-agnostic defenses, recovering up to 7.1 accuracy points over a no-defense baseline on highly-corrupted videos while maintaining competitive performance on clean/near-clean data.

## 1 INTRODUCTION

Video is becoming an increasingly common data modality, with applications in online conferencing (Jansen et al., 2018), autonomous vehicles (Chen et al., 2018; Bojarski et al., 2016), action recognition (Kuehne et al., 2011; Soomro et al., 2012; Carreira & Zisserman, 2017), and event detection (Fu et al., 2019; Gaidon et al., 2013). Recent work in computer vision has studied the robustness of machine learning (ML) models to pixel-space corruptions such as adversarial examples (Madry et al., 2017; Carlini & Wagner, 2016; Goodfellow et al., 2014; Moosavi-Dezfooli et al., 2016; Szegedy et al., 2013). However, in the real world, videos are susceptible to corruptions beyond the pixels, such as bit-level *network and file corruptions* (Fig. 1).

Bit-level corruptions are generally non-adversarial and arise in the real world from network congestion (Mushtaq & Mellouk, 2017), *i.e.,* transmission problems in video conferencing, or hardware errors during storage (Sivathanu et al., 2005; Zhang et al., 2010). These corruptions spuriously modify bits in a video file, leading to data loss or visual distortion (*e.g.* object duplication, noisy patches, or freeze-frames). Although some previous work in computer vision robustness has been compresssion-aware, e.g. studying the effects of JPEG compression (Aydemir et al., 2018; Dziugaite et al., 2016; Das et al., 2018) or changing bitrates (Guo et al., 2017b), direct corruptions to the bit representations of videos remain underexplored.

In this work, we take a first step in understanding video model robustness to naturally-occurring levels of network and file corruptions, and explore baseline defenses to these corruptions. To this end, we simulate a wide range of corruption severity levels consistent with empirical studies of real-world corruptions (Schroeder et al., 2016; Hu & Zhang, 2018), and explore their effects on video artifacts and model performance. We evaluate model performance on two action recognition datasets (Kuehne et al., 2011; Soomro et al., 2012) and one multi-object tracking dataset (Leal-Taixé et al., 2015) under simulated network packet loss and random bit-flip file corruptions. Our experiments demonstrate that model performance begins to degrade when packet loss rates exceed 1%, or when the proportion of randomly-flipped bits in a file exceeds $10^{-6}$, and ultimately drops by up to 77.1% under the most severe corruption levels.

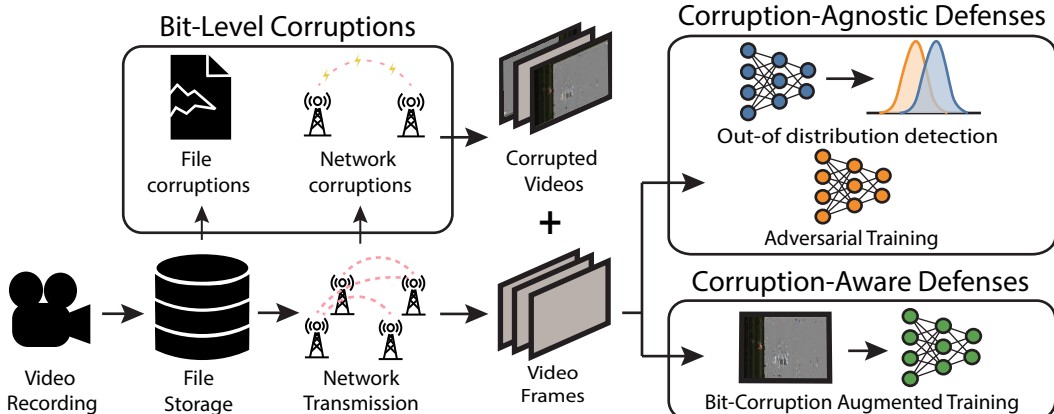

Figure 1: An overview of the video machine learning pipeline under bit-level corruptions, which can arise due to file storage and network transmission. We explore various defenses against network and file corruptions.

We then focus on exploring two categories of defenses against network and file corruptions: *corruption-agnostic* methods without knowledge of bit-level corruptions, and *corruption-aware* methods that exploit knowledge of bit-level corruptions to enforce model invariance to network and file corruptions. For corruption-agnostic defenses, we evaluate out-of-distribution (OOD) detection (Hendrycks & Gimpel, 2016; Liang et al., 2018) and adversarial training (Goodfellow et al., 2014) as baselines. Our findings suggest that corruption-agnostic defenses have limited effectiveness, especially at low corruption levels; for example, under adversarial training, model performance drops by up to 11.3 points on corrupted data, and 8.6 points on clean data, compared to the no-defense baseline.

In response, we propose **B**it-corruption **A**ugmented **T**raining (**BAT**), a corruption-aware baseline. BAT augments data with corrupted video files from randomly chosen bit-level corruptions during training. Experiments show that BAT recovers up to 7.1 points over a no-defense baseline on highly-corrupted videos, while maintaining competitive performance on clean/near-clean data. This suggests that the noise structure of bit-level corruptions is fundamentally different from that of adversarial noise, and that incorporating knowledge of bit-level corruptions is crucial to model robustness against network and file corruptions. Lastly, to better understand the effect of augmenting with different corruptions, we explore variants of BAT that sample augmentations from different subsets of simulated corruptions. Interestingly, we find that augmenting with high levels of network and file corruption is key to improving model robustness, resulting in an accuracy gain of up to 10.1 points over the no-defense baseline. We conclude that BAT is a promising starting point for robustness to bit-level corruptions in video machine learning. Our results motivate future studies to understand and defend against bit-level corruption in ML robustness.

## 2 BIT-LEVEL VIDEO CORRUPTIONS

In this section, we first provide a brief overview on video encoding to provide intuition about the effects of network and file corruptions (Section 2.1), and then describe how we simulate a naturally-occurring range of network and file corruptions (Section 2.2).

### 2.1 VIDEO ENCODING

We present a simplified explanation of video encoding, using the H.264 codec as an example (Richardson, 2003). Let $\mathcal{X} = \mathbb{R}^{H \times W \times C \times T}$ be the space of possible videos. An uncompressed video $\mathbf{x} \in \mathcal{X}$ consists of a sequence of frames. These frames are often redundant (i.e., two neighboring frames in a video are likely to look similar), so the H.264 codec stores only a small fraction of frames (known as `I-frames`) and encodes the remaining frames as differences from `I-frames`. To further save space, the frames go through a series of compression steps like frequency space conversion, quantization, and entropy encoding, much like JPEG compression. We notate the encoding process as the function `Enc(x)`, which is a mapping from pixel-space to bit-space. The corresponding decoding function is denoted as `Dec(·)`.

The H.264 video codec is also equipped with error resilience and concealment techniques for repairing damaged data. For example, the codec might use uncorrupted portions of a frame to decode a corrupted frame (Arbitrary Slice Order/Flexible Macroblock Order), or store redundant portions of a frame for the same purpose. However, bit-level network/file corruptions can destroy particular information such that these methods are insufficient to restore visual quality; for example, `I-frames` are important references for decoding other frames, so corruptions to them often result in propagating noise patches or smearing artifacts throughout the video.

## 2.2 SIMULATING BIT-LEVEL CORRUPTIONS

**Network corruptions.** We simulate *packet loss*, a network corruption where data packets are permanently lost during transmission (Fig. 2), which can result in visual quality drops in video streaming applications (Hu & Zhang, 2018; Adeyemi-Ejeye et al., 2017). We simulate packet loss by streaming videos using the Real-Time Streaming Protocol (RTSP) with UDP, dropping video packets randomly using the Mahimahi network emulator (Netravali et al., 2015). This setup targets real-time applications like video-conferencing that use UDP due to latency requirements.

**File corruptions.** We simulate *random corruptions*, a file corruption where random bits in a video file are erroneously flipped (Fig. 2). Such bit-flips occur as the result of hardware integrity issues like bus errors (Zhang et al., 2010), malicious software (Sivathanu et al., 2005), or cosmic rays (O'Gorman et al., 1996). Despite redundancy mechanisms in modern file systems, many such corruptions are unrecoverable due to unsafe fault-handling, or can persist undetected (Ganesan et al., 2017). In Appendix B, we additionally explore *contiguous corruptions*, another real-world bit-corruption pattern in which a contiguous segment of a file bitstream is replaced with random bits.

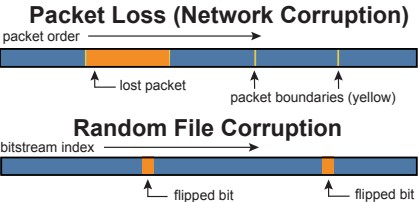

Figure 2: Depiction of corrupted bits (orange) affected by packet loss (top) and random file corruption (bottom).

**Corruption Levels.** Our primary interest is in simulating naturally-occurring levels of bit-level corruptions. Thus, we choose ranges for $p$ based on real-world empirical studies of packet loss and file corruption. Notationally, we describe corruptions with a parameter $p \in [0, 1]$, the expected proportion of the video file corrupted. For packet loss, each packet is dropped with probability $p$. For random corruptions, we flip each bit with probability $p$. Real-world network traces have yielded packet loss rates from 0.31% to 12.38% (Hu & Zhang, 2018), so we simulate packet loss rates from 0.01% ($p = 10^{-4}$) up to 20% ($p = 0.2$). On the file corruptions side, studies of flash memory have found median bit-error rates ranging from $6.0 \times 10^{-10}$ to $1.0 \times 10^{-5}$, with 99th percentile bit error rate up to $1.2 \times 10^{-4}$ (Schroeder et al., 2016), so we simulate file corruption rates from $p = 10^{-8}$ up to $p = 10^{-4}$. We provide more discussion of our chosen parameter settings in Appendix F.

## 3 BASELINE DEFENSES FOR BIT-LEVEL CORRUPTIONS

We explore three defenses as baselines for mitigating the impact of network and file corruptions. Broadly, these methods can be categorized as *corruption-agnostic*, which make no assumptions about network and file corruptions, or *corruption-aware*, which explicitly incorporate knowledge of network and file corruptions. As corruption-agnostic baselines, we evaluate out-of-distribution (OOD) detection and adversarial training (AT). We propose bit-corruption augmented training (BAT) as a corruption-aware baseline.

## 3.1 CORRUPTION-AGNOSTIC DEFENSES

Network and file corruptions often result in severe visual artifacts (see Section 4.2 for some examples). As a result, these corrupted videos can be viewed as out-of-distribution examples, particularly for large amounts of corruption. In such cases, a natural defense for pre-trained video models is OOD detection, a test-time technique that attempts to identify out-of-distribution videos and filters them out. This method requires no model retraining, and makes zero assumptions about the noise structure.

As an overview of OOD detection, we consider a ML model $f(\mathbf{x}; \theta) : \mathbb{R}^{H \times W \times C \times T} \rightarrow \mathbb{R}^K$ with parameters $\theta$, which are optimized under categorical cross-entropy loss:

$$\min_{\theta} \sum_{(\mathbf{x},y) \sim P} \ell(F_\theta(\mathbf{x}), y),$$

where $F_\theta(\mathbf{x})$ is the softmax output. We use ODIN (Liang et al., 2018), a standard OOD detection method that uses calibrated softmax score $S_\theta(\mathbf{x})$ as an indicator score for flagging OOD examples:

$$S_\theta(\mathbf{x}) = \max_j \frac{e^{f_j(\mathbf{x};\theta)/T}}{\sum_i e^{f_i(\mathbf{x};\theta)/T}},$$

where $T$ is a temperature scaling parameter (Guo et al., 2017a), and $f_i(\mathbf{x}; \theta)$ is the logit value for class $i$. The calibrated score $S_\theta(\mathbf{x})$ is then compared against a preset threshold (often 95% TPR) computed on in-distribution data; $\mathbf{x}$ is flagged as OOD if it falls below the threshold and is marked as in-distribution otherwise. The idea is that corrupted videos yield lower average calibrated scores than clean videos.

**Adversarial Training (AT).** Adversarial training (Goodfellow et al., 2014) is a common technique that makes models robust to small adversarial perturbations. Adversarial training optimizes

$$\min_{\theta} \sum_{(\mathbf{x},y) \sim P} \max_{||\delta||_\infty \leqslant \varepsilon} \ell(F_\theta(\mathbf{x} + \delta), y),$$

where $\varepsilon$ is the perturbation budget, or maximum permissible $L_\infty$-norm of perturbation $\delta$. We use the training scheme proposed in Wong et al. (2020), where an initial $\delta$ is randomly chosen from the $(L_\infty, \varepsilon)$-neighborhood of a video $\mathbf{x}$. Then, the above objective is iteratively optimized for one step. This method requires model retraining, and assumes that the noise structure of test-time input is adversarial.

## 3.2 CORRUPTION-AWARE DEFENSES

**Bit-corruption Augmented Training (BAT).** We consider a corruption-aware baseline defense that exploits knowledge of bit-level corruptions to enforce invariance to network and file corruptions. In particular, we propose *Bit-corruption Augmented Training* (BAT), which regularizes the video model by augmenting training data with examples from randomly chosen bit-level corruptions:

$$\min_{\theta} \sum_{(\mathbf{x},y) \sim P, \tau \in C} Z\ell(F_\theta((\texttt{Dec} \circ \tau \circ \texttt{Enc})(\mathbf{x})), y) + (1 - Z)\ell(F_\theta(\mathbf{x}), y),$$

where $C$ is the set of possible network and file corruptions, $\tau$ is a randomly chosen corruption for a particular video $\mathbf{x}$ applied with probability 0.5 (shown with indicator variable $Z \in \texttt{Bernoulli(0.5)}$). $\texttt{Enc}$ and $\texttt{Dec}$ denote the video encoding and decoding transformations, respectively, and $\circ$ denotes function composition. This method assumes prior knowledge about the nature of network and file corruptions, but not the exact corruption parameters for test-time videos. To better understand the effect of augmenting with different corruptions, we also consider variants of BAT that augment with a subset of corruption parameters.

## 4 EFFECT OF BIT-LEVEL CORRUPTIONS ON VIDEO MODELS

We provide an overview of our datasets and models in Section 4.1. In Section 4.2, we report the impact of bit-level corruptions on video model performance. We then report the performance of pre-existing corruption-agnostic defenses (Section 4.3), as well as Bit-corruption Augmented Training (BAT) as a corruption-aware defense (Section 4.4), finding that BAT outperforms the corruption-agnostic defenses. We also discuss variants of BAT to study the effect of training with particular subsets of the available corruptions.

## 4.1 DATASETS AND MODELS

**HMDB51.** The Human Motion DataBase (HMDB51) (Kuehne et al., 2011) is a 51-class action recognition benchmark comprised of movie scenes and web videos. We use a pre-trained 3D-Resnet18 architecture fine-tuned on HMDB51 split 1 (3520 train, 1530 test videos), using a standard

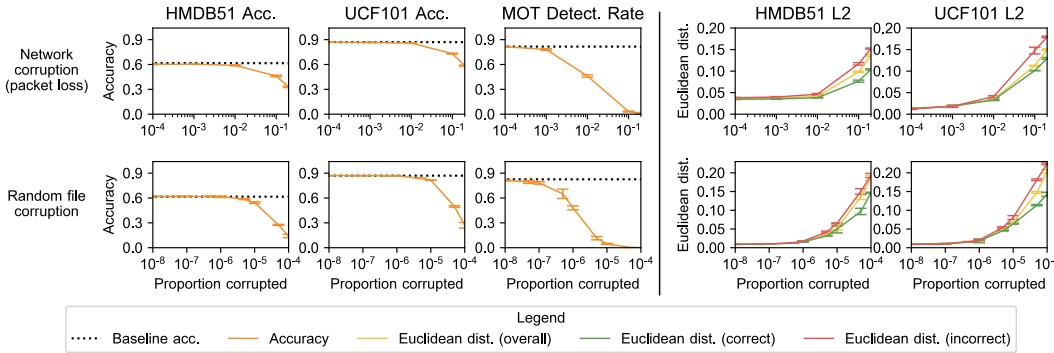

Figure 3: The impact of various file corruption types on model performance (left); average pixel-space Euclidean distance measurements (right) on incorrectly classified examples (red), correctly classified examples (green), and the entire dataset (yellow) organized by corruption proportion and type (right).

training setup (Kataoka et al., 2020). During evaluation, we split the input clip into contiguous 16-frame segments, outputting the action class with the highest probability averaged over all segments. We report top-1 accuracy.

**UCF101.** UCF101 (Soomro et al., 2012) is a 101-class action recognition benchmark comprised of web videos. Our setup is identical to that of HMDB51: we fine-tune a pre-trained 3D-Resnet18 architecture on UCF101 split 1 (9537 train, 3783 test videos) using the setup of Kataoka et al. (2020). We report top-1 accuracy.

**MOT15.** MOT15 (Leal-Taixé et al., 2015) is a multi-object tracking dataset. We use a DLANet-34 (Deep Layer Aggregation) architecture (Zhang et al., 2020) pre-trained on MOT20 (Dendorfer et al., 2020). We evaluate on a subset of the *train* split of MOT15 (6 clips, 12885 objects) following the setup in Zhang et al. (2020). We report object detection rate ($1 - FPR$), representing the proportion of objects correctly detected.

## 4.2 Effects of Bit-level Corruptions

**Effects of bit-level corruptions on video model performance.** We evaluate model performance on playable[1] videos under bit-level corruptions, finding that model accuracy drops substantially as corruption level increases (Fig. 3). On HMDB51 and UCF101, for random file corruptions, accuracy drops under packet loss for $p > 0.01$ and for random corruptions for $p > 10^{-6}$. Under packet loss, the performance drops reach 46.5% on HMDB51 and 32.6% on UCF101. For random file corruptions, the drop reaches 68.9% on HMDB51 and 77.1% on UCF101. On multi-object tracking, object detection rate drops to near 0 for all types of corruption. Notably, performance declines start at lower corruption levels for this task than for action recognition, starting at packet loss greater than $p = 10^{-3}$ and random corruption rates exceeding $10^{-7}$. More details on multi-object tracking performance are provided in Appendix E.

**Pixel-space effects of bit-level corruptions.** We visualize some corrupted videos to better understand the qualitative effects of bit-level corruptions. In general, misclassified videos are more visually distorted than correctly classified videos under corruption. We see this in Fig. 4: the model prediction changes as visual artifacts worsen and corruption level increases. Furthermore, under the same corruption type and proportion, visual artifacts correlate with erroneous model predictions, as seen in Fig. 5. In the leftmost pair of clips in Fig. 5, under packet loss ($p = 0.1$), we see noisy patches in the Kayaking clip (incorrect), while the sky, grass, and person in the GolfSwing example (correct) are not distorted. Similar patterns hold for random corruption (Fig. 5, right). On MOT15, more distorted videos yield more false positive and false negative object detections. For visualizations of corrupted videos in MOT15, see Appendix E.

---

[1]Video files can become unplayable if file metadata is corrupted. Metrics are calculated on playable videos only: if a video is unplayable, the model does not output a prediction at all. We evaluate accuracy purely based on outputted model predictions. Appendix F gives a probabilistic overview of the effect of bit-level corruptions on video playability.

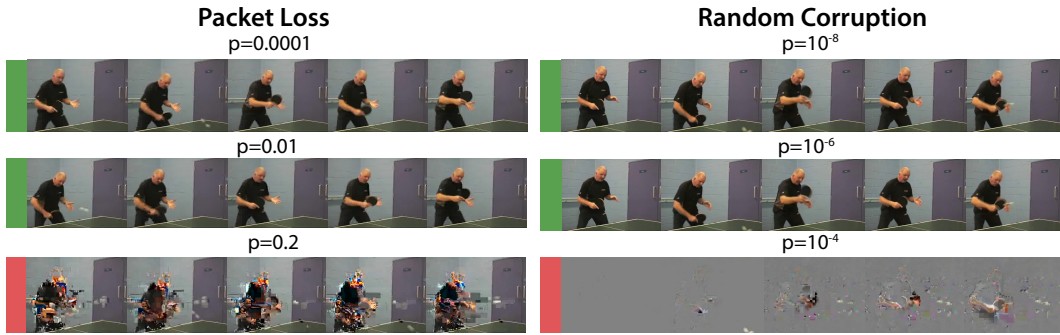

Figure 4: Frames from a clip (class `TableTennisShot`) at varying corruption levels. Color shows correct vs. incorrect classification. Videos become visually more distorted as the corruption level increases.

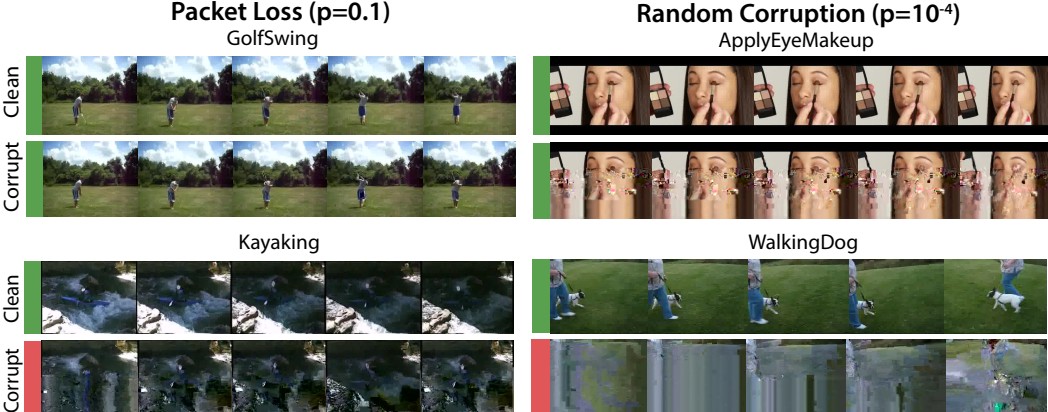

Figure 5: Frames from correctly and incorrectly classified examples, with both clean (1st and 3rd rows) and corrupted (2nd and 4th rows) versions shown. Each pair of corrupted clips were subjected to the **same** corruption proportion $p$ and mode (packet loss and random corruption).

To quantify this visual distortion, we compute pixel-space Euclidean distance ($L_2$ norm between pixels in corrupted vs. clean video clips). Fig. 3 shows the average $L_2$ distance between corrupted and clean clips on the entire dataset (yellow), correctly classified examples (green), and incorrectly classified examples (red). Consistent with our qualitative analysis, incorrect examples have higher $L_2$ distance on average than correct examples. Under packet loss, incorrect examples are up to $1.52$ times more perturbed than correct examples on average (HMDB51, network corruption, top left). For random corruptions, incorrect examples are up to $1.57$ more perturbed on average (UCF101, random corruption, bottom right). This strongly suggests that the visual severity of a network and file-corrupted video adversely affects model performance, as seen in Fig. 3.

### 4.3 CORRUPTION-AGNOSTIC DEFENSES FOR BIT-LEVEL CORRUPTIONS

**ODIN (Out-of-distribution Detection).** As observed in Section 4.2, videos with larger pixel-space distortions are more likely to be misclassified, motivating OOD detection as a baseline for filtering out corrupted videos. This defense does not require modifying model parameters, and can be applied to any pre-trained model. Specifically, we use a HMDB51 clean test set as in-distribution data, and a corrupted version of the same split as OOD data. During inference, we apply ODIN (Liang et al., 2018) with $T = 100$ to corrupted datasets as a filter, setting the score threshold to the 5th percentile of calibrated scores on the in-distribution set (95% TPR).

We evaluate the effectiveness of ODIN based on its ability to successfully filter out corrupted videos. Figure 6 shows AUROC for whether ODIN can successfully filter out corrupted videos, as well as model accuracy on the videos labeled as in-distribution (clean). We observe that ODIN is best able to detect corrupted videos at high levels of corruption, with AUROC reaching 98.5% for the highest levels of random corruption. ODIN is less effective at discriminating packet loss, with AUROC only reaching 76.6% at the highest levels of corruption. At low levels of corruption, ODIN discards many clean/near-clean examples, as AUROC degrades to 0.5. This is consistent with Fig. 3, which

Table 1: Robust accuracy on readable videos in the HMDB51 action recognition classification task (%) of no defense, adversarial training (AT), and bit-corruption augmented training (BAT) as defenses against network corruptions (top) and random file corruption (middle), with aggregate statistics (bottom).

| Corruption type | $p$ | No defense | AT ($\varepsilon = 2/255$) | BAT | BAT (Low) | BAT (High) | BAT (Oracle) |
|---|---|---|---|---|---|---|---|
| **Packet loss** | 0.2 | 32.8 | 27.0 | 39.9 | 32.4 | **41.8** | 32.0 |
| | 0.1 | 47.1 | 35.6 | **49.1** | 44.0 | 48.5 | 42.8 |
| | 0.01 | **59.0** | 48.3 | 58.3 | 57.4 | 57.7 | 58.4 |
| | $10^{-3}$ | **60.6** | 49.3 | 58.2 | 57.9 | 58.1 | 57.6 |
| | $10^{-4}$ | **60.0** | 49.4 | 58.9 | 58.4 | 58.7 | 58.0 |
| **Random file corruption** | $10^{-4}$ | 14.1 | 13.2 | 15.0 | 13.0 | 24.2 | **27.3** |
| | $5 \times 10^{-5}$ | 27.6 | 28.4 | 33.7 | 25.7 | 36.8 | **40.8** |
| | $10^{-5}$ | 54.5 | 48.0 | 55.6 | 52.8 | **55.7** | 54.1 |
| | $5 \times 10^{-6}$ | **58.7** | 51.3 | 56.3 | 56.3 | 57.7 | 56.7 |
| | $10^{-6}$ | **61.5** | 53.3 | 58.9 | 58.4 | 59.1 | 57.5 |
| | $5 \times 10^{-7}$ | 62.0[a] | 53.5 | 59.1 | 58.4 | 59.1 | 57.7 |
| | $10^{-7}$ | **61.9** | 53.5 | 59.1 | 58.5 | 59.0 | 58.5 |
| | $5 \times 10^{-8}$ | **61.9** | 53.5 | 59.1 | 58.4 | 58.9 | 58.6 |
| | $10^{-8}$ | **61.9** | 53.5 | 59.1 | 58.4 | 58.9 | 58.2 |
| **None (Standard Acc.)** | 0 | **61.6** | 53.0 | 59.5 | 59.0 | 59.0 | N/A |

[a] Accuracy on the no-defense baseline appears to surpass baseline at some corruption levels as the number of unplayable videos is non-zero at all levels of corruption tested.

suggests that the pixel-space gap between correctly classified and incorrectly classified videos narrows at low levels of corruption, meaning that corrupted and clean videos are more distributionally similar. In conclusion, ODIN shows some promise for filtering out severely corrupted videos, but cannot effectively discriminate clean from corrupted video at low levels of corruption. Thus, applying ODIN for OOD detection could result in good performance in filtering out highly corrupted data at the expense of discarding clean/lightly corrupted data. We additionally run evaluation on energy-based OOD (Liu et al., 2020), with almost identical results. Full results on OOD detection can be found in Appendix D.

**Adversarial Training.** Given the pixel-space proximity between clean and corrupted videos at low levels of corruption (Fig. 3, right), we explore adversarial training (AT), a family of methods for increasing model robustness to small adversarial perturbations (Goodfellow et al., 2014; Madry et al., 2017). We follow the method in Wong et al. (2020). The best results ($\varepsilon = 2/255$) are summarized in Table 1; larger $\varepsilon = 4/255, 8/255$ yield worse performance (Appendix C). Interestingly, it appears that AT offers little to no utility in increasing robustness to bit-level corruptions, underperforming a no-defense baseline both on corrupted data by up to 11.3 points (packet loss, $p = 10^{-3}$), and on clean data by 8.6 points. This suggests that the noise structure of bit-level corruptions is fundamentally

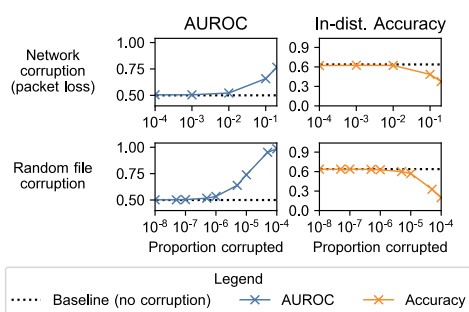

Figure 6: AUROC (left), in-distribution accuracy (right) of OOD detection under packet loss (top) and random file corruption (bottom).

different from that of adversarial noise, and that this form of AT may not be a viable defense against bit-level corruptions.

## 4.4 BIT-CORRUPTION AUGMENTED TRAINING

We propose **B**it-corruption **A**ugmented **T**raining (BAT) as a baseline corruption-aware defense that injects knowledge of file/network corruptions into video ML models. This technique augments training data with examples subjected to a randomly chosen network and file corruptions. We report the results of BAT, as well as a few variants, in Table 1. Overall, BAT shows promise, providing up to **7.1%** lift on highly-corrupted data (packet loss, $p = 0.2$) over a no-defense baseline. On

clean/near-clean data, BAT does only slightly worse than a no-defense baseline. This slight drop is likely attributable to the noise that BAT introduces into the model, since under BAT, we are now optimizing for accuracy not only on clean data, but also on corrupted data.

We study a few variants of BAT to determine which certain network and file corruptions are more important for increasing model robustness. First, we augment using two subsets of corruption levels: *low* levels of corruption, i.e. those that did not cause a significant performance drop on the baseline model, and *high* levels of corruption, i.e., corruption levels sufficient to cause significant performance drops in the baseline model[2]. Training on low levels of corruption (BAT-Low) decreases model robustness across almost all levels of corruptions, but training with high levels of corruption (BAT-High) results in an accuracy gain of up to **10.1**% (random file corruption, $p = 10^{-4}$) over the no-defense baseline—and outperforms vanilla BAT at almost all levels of corruption. As with vanilla BAT, the lift over the no-defense baseline is correlated with corruption proportion. This suggests that augmenting with highly corrupted videos in particular is key to increased robustness.

Lastly, we explore the effect of training and testing on the same corruption type and level (BAT-Oracle) as an oracle model with perfect knowledge of the corruption parameters the model will encounter. Though this assumption is unrealistic, BAT-Oracle provides insight into the utility of incorporating particular corruption levels within the BAT augmentation scheme. Interestingly, this technique lags behind BAT-High at high levels of corruption, and performs worse than BAT, BAT-High, and the no-defense baseline at low levels of corruption. This suggests that training on a variety of different corruptions is important for increasing robustness. Thus, BAT is a promising starting point for improving robustness to bit-level corruptions in video machine learning, especially for more severe corruptions. Our results motivate future studies of understanding and defending against bit-level corruptions.

## 5 RELATED WORK

**Robustness in computer vision.** Imagenet-C and -P (Hendrycks & Dietterich, 2019) are ImageNet (Deng et al., 2009) extensions for benchmarking image classifiers with non-adversarial corruptions. Hendrycks et al. (2020) studies model robustness to distributional shifts like occlusion and perspective shift. The effect of frequency-domain features on convolutional networks has also been studied (Yin et al., 2019; Tsuzuku & Sato, 2018). On the adversarial side, Szegedy et al. (2013) introduce the idea of adversarial examples in ML, while Goodfellow et al. (2014); Zheng et al. (2016); Madry et al. (2017) propose methods for creating and defending against adversarial input. Recent works also extend adversarial attacks to the video domain (Wang & Cherian, 2018; Wei et al., 2018; Inkawhich et al., 2018; Jiang et al., 2019; Naeh et al., 2020). Our work is not only an extension of previous image ML robustness studies to video, but also a study of network and file corruptions, which remain underexplored in the robustness literature.

**Defenses against corrupted data.** Augmenting data with speckle noise (Rusak et al., 2020), style transfer (Geirhos et al., 2018), mixtures of augmentations (AugMix) (Hendrycks et al., 2019), and distortions constructed from image-to-image networks (DeepAugment) (Hendrycks et al., 2020) have been shown to increase robustness to common corruptions like Imagenet-C. Other defenses include certificate-based methods (Raghunathan et al., 2018a;b) or self-supervision (Carmon et al., 2019). Non-adversarial perturbations can be interpreted as OOD data, for which supervised (Lee et al., 2018; Bahat & Shakhnarovich, 2018; Geifman & El-Yaniv, 2019; Gorbett & Blanchard, 2020) and semi-supervised OOD detection techniques (Liu et al., 2018) have been proposed. Video anomaly detection techniques are also explored in (Ben-Ari & Shwartz-Ziv, 2018; Gutoski et al., 2017). Adversarial training (Goodfellow et al., 2014; Madry et al., 2017) is a common defense to adversarial corruptions. We evaluate baseline non-adversarial and adversarial robustness techniques, and extend the augmentation-as-defense paradigm to bit-level corruptions.

**Bit-space aware robustness studies.** Previous work has studied the robustness of computer vision models under various aspects of image and video compression, such as compression rates or encoding schemes (Srinivasan et al., 2016; Seymour et al., 2007). On the network side, deep learning-based packet loss concealment (Lee & Chang, 2016; Lotfidereshgi & Gournay, 2018; Mohamed & Schuller, 2020) techniques aim to recover from packet loss by predicting lost information. We view

---

[2]Exact corruption parameters for low/high levels of corruption are described in Appendix F.

our work as complementary to previous efforts, extending robustness studies to explicitly examine the effect of network and file corruptions at the bit-level on video model performance.

## 6 CONCLUSION

In this paper, we take a first step in investigating the effect of network and file corruptions on video model robustness, finding that these corruptions significantly decrease performance on benchmark tasks like action recognition and multi-object tracking. We propose a corruption-aware defense baseline, Bit-corruption Augmented Training (BAT), which incorporates knowledge of network and file corruptions directly. We show that BAT can improve performance by up to 7.1% accuracy, and outperforms corruption-agnostic baselines (OOD detection and AT) at a wide range of corruption levels while perserving performance on clean data. Further experiments highlight the utility of augmenting with high levels of corruption in particular. This suggests that exploiting knowledge of network and file corruptions in model training is important for improving robustness, making BAT a compelling starting point for future studies of network and file corruptions and model robustness. We hope this work motivates future studies of defenses against network and file corruptions, as well as other real-world non-adversarial corruptions.

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

Table 2: Overall accuracy on all videos in the HMDB51 action recognition classification task (%) of no defense, adversarial training (AT), and bit-corruption augmented training (BAT) as defenses against network corruptions (top) and random file corruption (middle), with aggregate statistics (bottom).

| Corruption type | $p$ | No defense | AT ($\varepsilon = 2/255$) | BAT | BAT (Low) | BAT (High) | BAT (Oracle) |
|---|---|---|---|---|---|---|---|
| **Packet loss** | 0.2 | 27.2 | 25.6 | 33.0 | 32.4 | 35.4 | 25.0 |
| | 0.1 | 39.2 | 36.1 | 45.2 | 44.0 | 45.4 | 40.0 |
| | 0.01 | 51.3 | 51.8 | 57.3 | 57.1 | 57.4 | 56.3 |
| | $10^{-3}$ | 59.0 | 52.4 | 58.0 | 57.5 | 57.6 | 57.4 |
| | $10^{-4}$ | 59.3 | 52.7 | 58.6 | 58.2 | 58.5 | 57.6 |
| **Random file corruption** | $10^{-4}$ | 1.3 | 1.2 | 1.6 | 1.4 | 2.5 | 1.5 |
| | $5 \times 10^{-5}$ | 5.0 | 4.6 | 6.4 | 4.8 | 6.9 | 5.0 |
| | $10^{-5}$ | 31.0 | 26.4 | 33.0 | 32.5 | 34.3 | 32.8 |
| | $5 \times 10^{-6}$ | 43.4 | 35.4 | 43.3 | 43.5 | 44.6 | 43.9 |
| | $10^{-6}$ | 58.2 | 45.2 | 55.9 | 55.3 | 55.9 | 54.4 |
| | $5 \times 10^{-7}$ | 59.9 | 46.5 | 57.6 | 56.8 | 57.7 | 56.1 |
| | $10^{-7}$ | 61.4 | 47.1 | 58.8 | 58.3 | 58.8 | 58.3 |
| | $5 \times 10^{-8}$ | 61.6 | 47.6 | 59.1 | 58.4 | 58.8 | 58.0 |
| | $10^{-8}$ | 61.6 | 47.6 | 59.1 | 58.4 | 58.9 | 58.0 |
| **None (Standard Acc.)** | 0 | **61.6** | 53.0 | 59.5 | 59.0 | 59.0 | N/A |

Table 3: Model performance on HMDB51 with various defenses under contiguous file corruptions.

| Corruption type | $p$ | No defense | AT ($\varepsilon = 2/255$) | BAT | BAT (Low) | BAT (High) | BAT (Oracle) |
|---|---|---|---|---|---|---|---|
| **Contiguous file corruption** | 0.9 | 25.6 | 25.8 | 32.2 | 29.2 | 34.9 | **38.1** |
| | 0.75 | 41.4 | 37.2 | 44.1 | 41.2 | **46.0** | 44.6 |
| | 0.5 | 49.5 | 45.0 | 51.6 | 49.6 | **51.7** | 51.6 |
| | 0.25 | **55.0** | 47.8 | 54.9 | 52.9 | 54.6 | 53.9 |
| | 0.1 | **57.9** | 51.0 | 56.3 | 56.8 | 57.4 | 55.1 |
| | 0.01 | **61.2** | 52.0 | 59.5 | 57.8 | 59.2 | 58.5 |

## A    METRICS INCLUDING READABLE FILES

For completeness, we provide performance metrics here for accuracy on all videos, including unplayable videos. Unplayable videos are logged as incorrect. We notice similar trends as described in Table 3, in that BAT outperforms the no-defense baseline at high levels of corruption, and is close to the no-defense baseline at low corruption of accuracy. We also see similar results on the variants: BAT-High outperforms BAT at high levels of corruption, which suggests that augmenting with high levels of corruption is important for model robustness, while BAT-Oracle underperforms BAT in general, suggesting that augmenting with a wide range of corruptions positively influences model robustness. These are the same conclusions drawn in Section 4.4. As observed earlier, adversarial training generally hurts model performance.

## B    A STUDY ON CONTIGUOUS FILE CORRUPTIONS

We present a study on contiguous file corruptions. In contiguous corruptions, a segment of a video bitstream is replaced with random bits. These errors occur due to sudden temperature changes (thermal asperity), malware (Sivathanu et al., 2005), or misdirected data writes (Zhang et al., 2010). While these corruptions are plausible in the real world, corruptions do not yield significant performance drops or artifacts until over 10% of the file bitstream in all files are corrupted. For completeness, we include results on this type of corruption in Table 2, as well as in the remaining results in the Appendix.

## C  FULL RESULTS OF ADVERSARIAL TRAINING

In summary, adversarial training offers little to no utility in defending against network corruptions. In Table 4, we see that at all levels of corruption except for random corruption, $p = 5e - 5$ and contiguous corruption, $p = 0.9$, adversarially trained models are significantly less accurate than the normally trained model. We also notice that as $\varepsilon$ decreases (i.e. the adversary weakens), adversarial training accuracy tends to get closer to baseline accuracy. Interestingly, the gap between an adversarially trained model and the baseline model narrows as corruption proportion increases. Thus, this mode of adversarial training is an insufficient defense against network and file corruptions.

Table 4: Accuracy of adversarially trained models ($\varepsilon = 2/255, 4/255, 8/255$) against random file (top), contiguous file (middle), and network corruptions (bottom). **AT** = adversarial training via FGSM + random initialization.

|  | $p$ | No defenses | AT ($\varepsilon = 2/255$) | AT ($\varepsilon = 4/255$) | AT ($\varepsilon = 8/255$) |
|---|---|---|---|---|---|
| | 0.2 | **33.8** | 30.1 | 27.0 | 26.7 |
| | 0.1 | **45.6** | 38.6 | 35.6 | 34.7 |
| **Packet loss** | 0.01 | **59.5** | 52.1 | 48.3 | 46.0 |
| | $10^{-3}$ | **61.4** | 52.8 | 49.3 | 46.9 |
| | $10^{-4}$ | **60.5** | 52.9 | 49.4 | 47.3 |
| | $10^{-4}$ | **14.1** | 13.2 | 9.7 | 11.8 |
| | $5 \times 10^{-5}$ | 27.6 | **28.4** | 23.2 | 10.4 |
| | $10^{-5}$ | **54.5** | 48.0 | 43.8 | 23.2 |
| **Random file** | $5 \times 10^{-6}$ | **58.7** | 51.3 | 47.4 | 44.4 |
| **corruptions** | $10^{-6}$ | **61.5** | 53.3 | 49.2 | 46.5 |
| | $5 \times 10^{-7}$ | **62.0** | 53.5 | 49.4 | 46.8 |
| | $10^{-7}$ | **61.9** | 53.5 | 49.5 | 46.7 |
| | $5 \times 10^{-8}$ | **61.9** | 53.5 | 49.3 | 46.7 |
| | $10^{-8}$ | **61.9** | 53.5 | 49.3 | 46.7 |
| | 0.9 | 25.6 | 25.8 | **26.5** | 20.4 |
| | 0.75 | **41.4** | 37.2 | 37.4 | 35.9 |
| **Contiguous file** | 0.5 | **49.5** | 45.0 | 43.4 | 40.3 |
| **corruptions** | 0.25 | **55.0** | 47.8 | 44.2 | 44.4 |
| | 0.1 | **57.9** | 51.0 | 47.1 | 45.7 |
| | 0.01 | **61.2** | 52.0 | 48.1 | 46.9 |
| **No corruption** | 0 | **61.6** | 53.0 | 49.5 | 47.5 |

## D  FULL RESULTS OF OUT-OF-DISTRIBUTION DETECTION

### D.1  MACHINE LEARNING-BASED SOLUTIONS

Table 5 provides full results for our study of ODIN OOD detection as a defense for network and file corruptions. We report in-distribution accuracy (left) and AUROC (right). As seen in Fig. 6, AUROC is relatively high at the maximum levels of corruption (76.6% on packet loss, 98.5% on random file corruptions), but drops down to essentially random at low levels of corruption.

Table 6 provides full results for the energy-based OOD detection (Liu et al., 2020) framework as a defense for network and file corruptions. As a brief overview, similarly to ODIN, energy-based OOD detection produces a score given by the Helmholtz free energy function, which is

$$S_\theta(\mathbf{x}) = -T \cdot \sum_{j}^{K} e^{f_j(\mathbf{x};\theta)/T},$$

with $T$ and $f_j(\mathbf{x}; \theta)$ as defined for ODIN.

We report in-distribution accuracy (left) and AUROC (right). The setup is identical to ODIN, except we have $T = 1$ following the authors' specification. Results for energy-based OOD detection are very similar to that of ODIN, with AUROC as high as 77.0% and 98.7% on packet loss and random

Table 5: In-distribution accuracy and AUROC for ODIN OOD detection ($T = 100$), HMDB51.

| Corruption type | $p$ | In-dist. Accuracy | AUROC |
|---|---|---|---|
| **Packet loss** | 0.2 | 37.4 | 76.6 |
| | 0.1 | 48.5 | 65.8 |
| | 0.01 | 62.3 | 52.2 |
| | $10^{-3}$ | 62.5 | 50.5 |
| | $10^{-4}$ | 62.1 | 50.5 |
| **Random file corruption** | $10^{-4}$ | 19.8 | 98.5 |
| | $5 \times 10^{-5}$ | 32.8 | 95.2 |
| | $10^{-5}$ | 57.0 | 73.9 |
| | $5 \times 10^{-6}$ | 60.0 | 63.9 |
| | $10^{-6}$ | 62.9 | 53.4 |
| | $5 \times 10^{-7}$ | 63.5 | 51.7 |
| | $10^{-7}$ | 63.8 | 50.4 |
| | $5 \times 10^{-8}$ | 63.7 | 50.2 |
| | $10^{-8}$ | 63.7 | 50.2 |
| **Contiguous file corruption** | 0.9 | 35.5 | 96.2 |
| | 0.75 | 53.2 | 83.6 |
| | 0.5 | 52.9 | 71.1 |
| | 0.25 | 57.2 | 62.7 |
| | 0.1 | 60.1 | 58.7 |
| | 0.01 | 61.6 | 55.6 |
| **None** | 0 | 63.8 | 50.0 |

Table 6: In-distribution accuracy and AUROC for energy-based OOD detection, HMDB51.

| Corruption type | $p$ | In-dist. Accuracy | AUROC |
|---|---|---|---|
| **Packet loss** | 0.2 | 36.6 | 77.0 |
| | 0.1 | 47.3 | 66.3 |
| | 0.01 | 62.7 | 52.5 |
| | $10^{-3}$ | 62.4 | 50.7 |
| | $10^{-4}$ | 62.6 | 50.5 |
| **Random file corruption** | $10^{-4}$ | 9.1 | 98.7 |
| | $5 \times 10^{-5}$ | 31.8 | 95.8 |
| | $10^{-5}$ | 54.5 | 74.2 |
| | $5 \times 10^{-6}$ | 60.2 | 64.7 |
| | $10^{-6}$ | 63.5 | 53.5 |
| | $5 \times 10^{-7}$ | 63.4 | 51.8 |
| | $10^{-7}$ | 63.6 | 50.5 |
| | $5 \times 10^{-8}$ | 63.6 | 50.2 |
| | $10^{-8}$ | 62.0 | 50.3 |
| **Contiguous file corruption** | 0.9 | 36.6 | 96.2 |
| | 0.75 | 49.4 | 84.5 |
| | 0.5 | 52.9 | 71.1 |
| | 0.25 | 57.1 | 62.3 |
| | 0.1 | 58.3 | 58.8 |
| | 0.01 | 62.1 | 55.0 |
| **None** | 0 | 63.6 | 50.0 |

file corruptions, respectively. However, both techniques are near random at low levels of corruption. We provide a side-by-side comparison of ODIN and energy-based OOD detection in Figure **??**, with plots of AUROC and in-distribution accuracy under packet loss and random file corruptions.

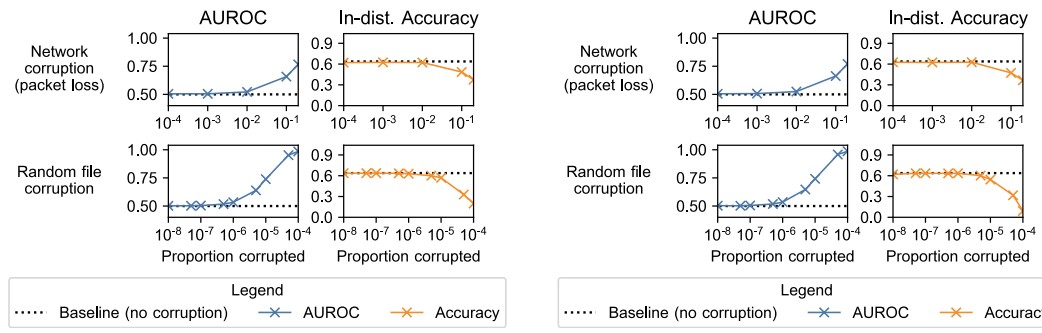

(a) AUROC and in-distribution accuracy of ODIN OOD detection, HMDB51.

(b) AUROC and in-distribution accuracy of energy-based OOD detection, HMDB51.

Figure 7: Comparison of ODIN and energy-based OOD detection methods.

Table 7: In-distribution accuracy and support for checksum-based corrupted video detection, HMDB51.

| Corruption type | $p$ | In-dist. Accuracy | Support (# not corrupted) |
|---|---|---|---|
| **Packet loss** | 0.2 | N/A | 0 |
| | 0.1 | N/A | 0 |
| | 0.01 | N/A | 0 |
| | $10^{-3}$ | N/A | 0 |
| | $10^{-4}$ | N/A | 0 |
| **Random file corruption** | $10^{-4}$ | N/A | 0 |
| | $5 \times 10^{-5}$ | N/A | 0 |
| | $10^{-5}$ | 70.0 | 10 |
| | $5 \times 10^{-6}$ | 66.7 | 42 |
| | $10^{-6}$ | 63.1 | 620 |
| | $5 \times 10^{-7}$ | 62.7 | 944 |
| | $10^{-7}$ | 62.3 | 1393 |
| | $5 \times 10^{-8}$ | 61.9 | 1508 |
| | $10^{-8}$ | 61.9 | 1508 |
| **None** | 0 | 61.6 | 1530 |

## D.2 CHECKSUM-BASED FILE INTEGRITY CHECKS

In Table 7, we provide results testing the effectiveness of checksum-based file integrity checks, a model-agnostic, machine learning-free technique for detecting file corruption. We flag a video as corrupted/out-of-distribution if the checksums of the corrupted vs. original video fail to match. We use the MD5, SHA1, and SHA256 checksums using the corresponding Linux utilities; however, all checksums produced identical detections.

Ultimately, this method discards a large quantity of clean or near-clean videos, meaning that its effectiveness is limited. In fact, even re-streaming a video through a link with *zero* packet loss results in checksums no longer being equal, which can happen if there are differences in the uplink and downlink codec parameters. Since checksums are merely binary indicators of corruption, and have no notion of how severe or recoverable a corruption is, this is a core limitation of this technique in the setting of model robustness to bit-level network/file corruptions.

# E MULTI-OBJECT TRACKING

We provide full results for our robustness study of multi-object tracking, with multi-object tracking, false positive count, and false negative count.

## E.1 ROBUSTNESS STUDY RESULTS

On the MOT task, for all modes of corruption, multi-object tracking accuracy (MOTA) drops (Fig. 9). For random and network corruptions, performance drops to 0, while for contiguous corruptions, performance drops by 160% to $-0.453$. The negative value is possible due to the form of the MOTA calculation:

$$MOTA = 1 - \frac{\sum_t FP_t + FN_t + SWITCH_t}{\sum_t OBJ_t} \tag{1}$$

where $FP_t$ is the number of false positive detections at frame/timestep $t$, $FN_t$ is the number of missed/false negative detections at timestep $t$, $SWITCH_t$ is the number of "identity switches" at time $t$ (i.e. one object is now classified as another object), and $OBJ_t$ is the total number of objects at time $t$. The negativity comes from the fact that $FP_t$ and $SWITCH_t$ are not bounded above. This is likely attributable to the false positive rate being much higher on the contiguous corruption experiments; empirically, the maximum average false positive count in these experiments is 7.41x greater than the maximum average number (8030, contiguous corruption, $p = 0.75$) of false positives in any other experiment (1083, packet loss, $p = 0.01$).

We also comment on the trends in the false positive and negative counts (center, right columns of Fig. 9). On each type of corruption, the number of false positives first increases, then decreases as corruption proportion increases. Similarly, the number of false negatives consistently increases. On average, there are overwhelmingly more false negatives than false positives.

To understand the patterns in false positive and false negative counts, we provide an illustrative visualization for packet loss in Fig. 8. In the ground truth example (Fig. 8a), the model detects three individuals, but in the example with packet loss $p = 0.01$ (Fig. 8b), we see multiple object duplication artifacts, resulting in a double detection of the rightmost individual (false positive). This signals a partial non-`I-frame` corruption, as the motion information fails to erase some parts of the old image, while the new location of the individuals is not updated correctly, resulting in the double images. However, as corruption proportion continues to increase, the video becomes so damaged that the model makes very few detections, hence the continual increase in the number of false negatives. This is shown in Fig. 8c, showing an example at packet loss $p = 0.1$, as the video becomes so distorted that two out of three of the individuals detected in the clean image are not detected at all. This signals `I-frame` destruction, as the image background is entirely grayed-out. The same visual trends hold on contiguous and random corruption.

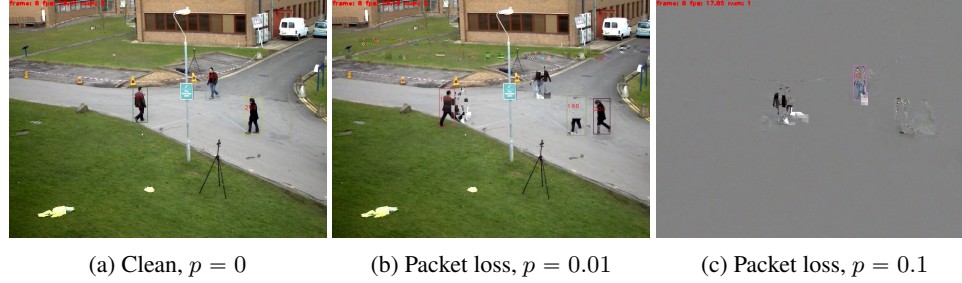

(a) Clean, $p = 0$       (b) Packet loss, $p = 0.01$      (c) Packet loss, $p = 0.1$

Figure 8: Frame #8 from the PETS09-S2L1 sequence, showing model predictions on clean data (left), $p = 0.01$ packet loss (center), and $p = 0.1$ packet loss (right).

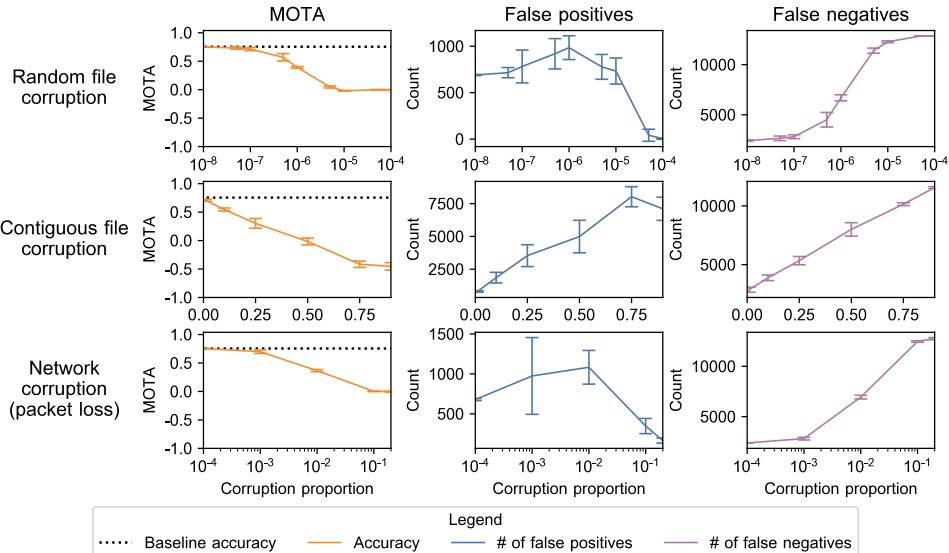

Figure 9: The impact of various file corruption types on model accuracy, false positive count, and false negative count for the multi-object tracking task. Error bars denote standard deviation over 5 random seeds.

## F ADDITIONAL EXPERIMENTAL DETAILS

### F.1 CORRUPTION SIMULATION PARAMETERS

**Random file corruptions.** We tested the effect of random corruptions on a semi-logarithmic scale, experimenting with $p = [1e{-}8, 5e{-}8, 1e{-}7, 5e{-}7, 1e{-}6, 5e{-}6, 1e{-}5, 5e{-}5, 1e{-}4]$. We chose this scale since corruption proportions greater than $p = 1e - 4$ yielded too few playable examples to get stable estimations of the accuracy and average pixel-space Euclidean distance. To randomly corrupt a video, we convert the video file into a raw bytearray and flip each bit independently with probability $p$. For BAT variants, we define our cutoff for low levels of corruption as $p \leqslant 10^{-6}$, and corruption as *high* otherwise.

**Contiguous file corruptions.** We tested the effect of contiguous corruptions at $p = [0.01, 0.1, 0.25, 0.5, 0.75, 0.9]$. We chose this scale since the effect of corruption proportion on model accuracy and average pixel-space Euclidean distance was more visually apparent on this scale than on a logarithmic scale. To apply contiguous corruption, we convert the video file into a raw bytearray and choose a random starting bit from between 0 and $\lfloor (1 - p) \cdot L \rfloor$, where $L$ is the length of the video. For BAT variants, we define our cutoff for low levels of corruption as $p \leqslant 0.1$, and corruptions as *high* otherwise.

**Packet loss.** We tested the effect of packet loss rates $p = [0.0001, 0.001, 0.01, 0.1, 0.2]$. This parameter range yielded a wide variety of artifacts and allowed us to observe the accuracy decline of pre-trained video models. We stream videos using `ffmpeg` using the Real-Time Streaming Protocol (RTSP) over the User Datagram Protocol (UDP), and simulate packet loss by streaming within `LossShell` in the `mahimahi` network simulation program (Netravali et al., 2015). Each packet is dropped independently with probability $p$. For BAT variants, we define our cutoff for low levels of corruption as $p \leqslant 0.01$, and corruption as *high* otherwise.

**Probabilistic perspective on bit-level corruptions.** We provide a brief probabilistic perspective on the chance of corruptions making the video unplayable by hitting a critical section of the video file bitstream. For a video of $P$ packets, the probability a video becomes unreadable is $1 - (1 - M)^{P \cdot p}$, where $M$ is the fraction of video composed of critical metadata. The probability of a corruption hitting a critical section due to a random corruption is $1 - (1 - M)^{B \cdot p}$, where $B$ is the number of bits in the video ($P \ll B$). For contiguous corruptions, making the simplifying assumption that critical sections of the video are contiguous, the expression is $\lfloor M/(1 - p) \rfloor$, which is close to linear outside the neighborhood of $p = 1$ for sufficiently small $M$. In practice, $M$ varies by video container

format (*e.g.* AVI, MP4) and with the size of the video file. Replacing $M$ with the proportion of bits in the video that results in distortions when corrupted, we can derive analogous expressions for the probability of a visible corruption.

## F.2 MODEL ARCHITECTURE AND CONFIGURATION

**Training.** For action recognition, follow the fine-tuning setup in Kataoka et al. (2020). We obtained the pre-trained model from the 3D-ResNets-PyTorch GitHub repository (`https://github.com/kenshohara/3D-ResNets-PyTorch`), using the model trained for 200 epochs on Kinetics-700 (Carreira et al., 2019) and MiT (Monfort et al., 2018) (`r3d18_KM_200ep.pth`). We fine-tune on split 1 of HMDB51 and UCF101, sampling a random 16-frame segment from each video. If the video is shorter than 16 frames, we loop the video to the required length. All videos are resized to 112 pixel frames. We also apply with a probability 0.5 a random crop across all frames of the segment with crop scales $\{1, \frac{1}{2^{\frac{1}{4}}}, \frac{1}{\sqrt{2}}, \frac{1}{2^{\frac{3}{4}}}, \frac{1}{2}\}$, where a crop scale of 1 means that the crop has the same width and height, a crop scale of $\frac{1}{2}$ means that the shorter side is $\frac{1}{2}$ the length of the longer side, and likewise for the other crop scales. We flip each clip with probability 0.5 and subtract the mean values of ActivityNet from each channel (RGB: $[114.7748, 107.7354, 99.4755]$). We train for 50 epochs using stochastic gradient descent with a learning rate $1e-3$, momentum $0.9$, and weight decay of $1e-5$. We do not use early stopping; we reduce the learning rate by a factor of 10 if validation loss does not improve for 10 epochs, training for a maximum of 50 epochs. We use the same training parameters for adversarial training and corruption-augmented training.

We do not perform further training on multi-object tracking.

**Evaluation.** For action recognition, we split the clip into contiguous non-overlapping 16-frame segments, matching the length of segments used during training, and evaluate the model on each segment. We also subtract the mean values of ActivityNet from each channel (RGB: $[114.7748, 107.7354, 99.4755]$) during evaluation for consistency with training. If the final video segment is shorter than 16 frames, we loop it until it reaches the requisite 16 frames. For our prediction, we output the class with the highest probability averaged over all segments.

For multi-object tracking, we simply pass in the corrupted versions of MOT15-*train* to the preset evaluation script for MOT15, which can be found at `https://github.com/ifzhang/FairMOT`.

**Hardware.** Training and evaluation was done on one NVIDIA Tesla P100 GPU with 16GB VRAM.

## F.3 CALCULATING PIXEL-SPACE EUCLIDEAN DISTANCE

For completeness, we explicitly formulate our calculation of pixel-space Euclidean distance. A video can be interpreted as a four-dimensional tensor, with time, channel (representing colors), height, and width dimensions, which we can notate as $C, T, W, H$, respectively. In the HMDB51 and UCF101 datasets, the channel dimension has size 3, corresponding to the RGB values of each pixel, which we represent as a float between 0 and 1 inclusive.

Average pixel-space Euclidean distance measures the amount each pixel at a point in time and space changes on average. This is calculated as the pixel-wise $L_2$ norm in the spatial and temporal dimensions averaged over all pixels in the video tensor. As an example, suppose that we have two video tensors, $V_0$ and $V_1$, each with dimensions $(C \times T \times W \times H)$. Consistent with Section F.2, in our experiments, $C = 3$ and $W = H = 112$. The number of frames in a video $T$ varies by video. In the case that durations $T_0$ and $T_1$ for videos $V_0$ and $V_1$, respectively, are different, we take length $T' = \min(T_0, T_1)$ and truncate each video in the time dimension to have dimension $(C \times T' \times W \times H)$. Let $D = V_0 - V_1$, the element-wise difference; then the pixel-space Euclidean distance is

$$\frac{1}{\sqrt{3} \cdot T' \cdot W \cdot H} \sum_{t=1}^{T} \sum_{w=1}^{W} \sum_{h=1}^{H} \|D_{twh}\|_2, \tag{2}$$

where $D_{twh}$ is the pixel at the location specified by $(w, h)$ at time $t$. Note the $\sqrt{3}$ in the denominator of the left-most term of Eq. 2. This is a normalization constant that ensures the pixel-space Euclidean distance between two video tensors stays between 0 and 1 inclusive.

