# OpenReview forum: "Beyond the Pixels: Exploring the Effects of Bit-Level Network and File Corruptions on Video Model Robustness"
_ICLR.cc/2021/Conference — Reject_

### Official Review · AnonReviewer3 · 2020-10-22
**Extensive experiments but limited novelty**

**Rating:** 4
**Confidence:** 3

**Review:**

Post-rebuttal: The rebuttal partly addresses my concerns, so I would like to change my score to 4.

This paper simulates network and file corruptions at multiple corruption levels, and explore corruption-agnostic and corruption-aware defenses. The presented Bit-corruption Augmented Training enhances the robustness of the video machine learning models. Experimental results show the effectiveness.

Pros:
The experiments are very extensive. The effectiveness of the presented method compared with the corruption-agnostic is clearly validated.

Cons:
1.      The novelty is limited. Considering the attack in the encoding and decoding process have been studied, and sophisticated methods [1] have been proposed. Compared with [1], I wonder about the advantage of the proposed method.
2.      The experimental results are not convincing. Baselines such as [1] is not included in the experimental results for fair comparison. I think the baselines in the manuscript is too weak.
3.      I think the idea is too ad-hoc, which just generates adversarial samples in the space of encoder and decoder instead of the space of the classifier. I think the difference is not significant.

[1] Rakin, A. S., He, Z., & Fan, D. (2019). Bit-flip attack: Crushing neural network with progressive bit search. In Proceedings of the IEEE International Conference on Computer Vision (pp. 1211-1220).

---

> ### Author Response · Authors · 2020-11-18
> **[draft updated] Responding to concerns on comparisons and setup. Thank you for the feedback!**
>
> We thank R3 for their positive comments on the effectiveness of our method and the extensiveness of our experiments.
>
> **What about comparing to the bit-flip attack baseline [1]?**
> We thank R3 for pointing out this work. While that work is interesting, it pertains to an attack on the model weights based on flipping bits [1], while we study bit-level corruptions in the data files. Thus, while we share interests in bit-level corruptions, we are ultimately targeting a different problem. Nevertheless, we are encouraged to see that others find the study of bit-level corruptions interesting.
>
> **Is our simulation of corrupted examples too “ad-hoc?”**
> We simulated these network/file corruptions with naturally-occurring scenarios in mind; our simulation parameters align with real-world bit-error/packet loss rates [2, 3]; **we discuss this in Section 2.2** (pg. 2). Furthermore, video is transmitted in compressed format (encoder/decoder space). Since submission, we have put even more thought into this, and are aiming to simulate real-world network failure modes based on real-world network observations, such as those described in the Pantheon network link testbed [4]. We have emphasized that we are targeting naturally-occurring scenarios more explicitly in the current draft.
>
> [1] Rakin, Adnan Siraj, Zhezhi He, and Deliang Fan. "Bit-flip attack: Crushing neural network with progressive bit search." Proceedings of the IEEE International Conference on Computer Vision. 2019.
>
> [2] Hu, Zhiguo, and Qiqiang Zhang. "A new approach for packet loss measurement of video streaming and its application." Multimedia Tools and Applications 77.10 (2018): 11589-11608.
>
> [3] Schroeder, Bianca, Raghav Lagisetty, and Arif Merchant. "Flash reliability in production: The expected and the unexpected." 14th USENIX Conference on File and Storage Technologies (FAST 16). 2016.
>
> [4] Yan, Francis Y., et al. "Pantheon: the training ground for Internet congestion-control research." 2018 USENIX Annual Technical Conference (USENIX ATC 18). 2018.

---

### Official Review · AnonReviewer4 · 2020-10-28
**The paper needs to clearly define the threat model and compare with fault-tolerance methods**

**Rating:** 3
**Confidence:** 4

**Review:**

**Summary:**

The authors evaluate the effect of bit-level corruption, including network packet losses and bit corruptions, on video models such as action recognition and multi-object tracking. They found that the model performances drop significantly under severe corruption levels. To overcome this issue, they propose a defense method named Bit-corruption Augmented Training (BAT) to enhance the robustness of the model by embedding corrupted video samples in the training process. Results show that BAT is able to improve the model robustness over other methods such as Out-Of-Distribution (OOD) detection and Adversarial Training (AT).

**Strength:**

- Explores an interesting and new problem space for video model robustness.
- A range of realistic corruption rates was evaluated.

**Weakness:**
- The threat model for the defense is unclear.
- No comparison with naive fault tolerance methods at the network and video file level.
- The proposed BAT method is trivial and the performance is not good.
- Packet loss should not be categorized into data corruption.

**Detailed comments:**

This paper explores a topic that is both realistic and interesting for the increasingly popular video model applications. I appreciate that the authors evaluated their approach under realistic packet loss and bit-corruption rates that would occur in the physical world. Yet the paper has several points which I think significantly weaken its value. I will detail them as follows.

- The authors claim the proposed BAT method as well as OOD detection and AT as *defenses*. However, I can hardly find any evidence that the authors evaluated these methods under an adversarial setting, i.e., the paper does not assume the existence of an attacker. It seems to me that this paper is more on the side of fault tolerance than defenses for attacks.
- For detecting or correcting network packet losses and bit-corruptions in the files, a straightforward solution is applying existing network and memory/file-level fault-tolerance methods. For example, checksum or error-correction codes are two easily applicable solutions. The benefit of applying these methods is that they are model-agnostic and would not affect model performance under normal conditions.
- The BAT method proposed by the authors seems rather trivial to me. How is this different from data augmentation? In addition, the reported performance of BAT is not good. For example, would a 27.3% accuracy under 10^-4 random file corruption bring enough utility for the model?
- Lastly, I do not think it is a good idea to categorize packet losses as bit-level corruption since data loss is fundamentally different from data corruption. In other words, data corruption considers cases when data got changed but data loss only considers when data is missing.

---

> ### Author Response · Authors · 2020-11-18
> **[draft updated] Framing comments on threat model and evaluation of methods. Thank you for the constructive comments!**
>
> We thank R4 for their positive comment that this problem space is **new and interesting,** and for their recognition that we are targeting **realistic rates of data corruption.**
>
> **Why do we not assume the existence of an attacker?**
> We position ourselves as a robustness study, and designed our experiments with naturally-occurring scenarios in mind. Thus, our “adversaries” are naturally-occurring network failure conditions, and our experiments evaluate baseline defenses against such conditions. To support this framing, our simulation parameters align with real-world bit-error/packet loss rates [1, 2]; we have revised Section 2.2 of this draft  to convey this more explicitly (pg. 2).
>
> Furthermore, the three types of defenses we evaluate operate under increasingly strong but reasonable assumptions about the machine learning setup: for OOD detection, we assume we only have access to the test-time output (i.e. logits) of the model. For adversarial training, we assume retraining is feasible. Only BAT makes the assumption that we have prior knowledge of these corruptions. These assumptions have been explicitly addressed in the new draft. We have since been taking steps to study the effects of real-world bit-level corruptions on model robustness in more depth, such as network failure modes calibrated to real-world network traces like those simulated in the Pantheon network test bed [3].
>
> Ultimately, we see ourselves as taking a first step in understanding the robustness problem for video network/file corruptions. Thus, we are merely testing the effectiveness of reasonable robustness baselines in this new problem setting. We hope that our work provides a starting point for future studies of this problem.
>
> **How is BAT different from data augmentation, and is BAT “good enough?”**
> While existing methods have explored extensively on using data augmentation to improve model robustness, to our best knowledge, BAT is the first exploration of performing data augmentation **in the bit space** for improving model robustness. This is both new and non-trivial, as the transformation needs to go through additional encoding and decoding processes.
>
> We view BAT as a corruption-aware baseline for increasing model robustness to network/file corruptions. We are taking a first step in understanding the problem of model robustness in this setting, and to this end, we evaluate a variety of preexisting reasonable approaches to observe how they perform in this new setting. Furthermore, while considering the real-world utility of a model with 27.3% accuracy under 10^-4 bit error rate is important, it is not the primary goal of this paper. Even if the performance of BAT under high levels of corruption is subjectively low, it is our hope that future studies of this problem can build off of these baselines and our analyses.
>
>
> **On applying existing fault-tolerance methods:**
> We thank R4 for this suggestion. We tried this, and found that checksum-based integrity checks discarded large quantities of data even at low corruption levels. In particular, even streaming video with zero packet loss can result in a checksum violation due to small inconsistencies in the video encoding/decoding process as the video is streamed. We have added these results to Appendix D.2 (pg. 16).
>
>  The H.264 codec, which we use for all videos tested, already includes error correction measures, like adding redundant slices to the bitstream, as well as error concealment measures, such as slice/macroblock reordering at decoding time [4]. Even then, visually distorted videos as those in Figure 4 and 5 (pg. 6) still result. We have added this explanation to our overview of the H.264 codec in Section 2.1 (pg. 2).
>
>
> [1] Hu, Zhiguo, and Qiqiang Zhang. "A new approach for packet loss measurement of video streaming and its application." Multimedia Tools and Applications 77.10 (2018): 11589-11608.
>
> [2] Schroeder, Bianca, Raghav Lagisetty, and Arif Merchant. "Flash reliability in production: The expected and the unexpected." 14th USENIX Conference on File and Storage Technologies (FAST 16). 2016.
>
> [3] Yan, Francis Y., et al. "Pantheon: the training ground for Internet congestion-control research." 2018 USENIX Annual Technical Conference (USENIX ATC 18). 2018.
>
> [4] Richardson, Iain E. The H. 264 advanced video compression standard. John Wiley & Sons, 2011.

---

### Official Review · AnonReviewer1 · 2020-10-28
**an unfamiliar problem with familiar solutions**

**Rating:** 6
**Confidence:** 3

**Review:**

Summary:

This work investigates the problem of building robust video prediction models in the presence of signal corruption. The problem itself is not widely studied and experimental work like this one certainly opens some possibilities. The solution on the other hand is surprisingly simple and easy to implement. It serves the purpose of introducing the problem to a wider audience, and shed some light in different types of remedies.

Clarification:

1. What is the importance of solving this type of problem in the general video application setting? One would assume there are existing methods in signal processing without machine learning to tackle the issues well, especially when knowing the type of the noise and their corruption ratio in range. One would hope a section of related work on it and discuss the further reliance of a data driven approach.
2. The "No defense" baseline at some corruption levels are much better in accuracy. I'm not sure I understand the explanation given in Table 1.
3. What models are used to perform action recognition and tracking? Does the performance of BAT have anything to do with choice of models?

In  all, the problem itself seems worth a pursuit if stated with more context. The concern is that the solution is fairly specialized on the known noise types and their general range. One would argue that adding input noise has already been used in numerous previous work to improve the robustness of models.  The major issue, which is completely ignored, is that adding input noise could also hurt the model performance on clean inputs (assuming not all videos are corrupted). This is the aspect that is worth a discussion, a trade off that needs some illustration.

---

> ### Author Response · Authors · 2020-11-18
> **[draft updated] Some clarifications; on the clean/corrupted accuracy tradeoff of BAT. Thank you for the insightful questions and comments!**
>
> We thank R1 for their positive comments about the novelty of our problem setting and the simplicity of our proposed solution. We are also pleased to hear that R1 finds that our work can introduce the problem of machine learning robustness to bit-level network/file corruptions to a wider audience. We appreciate their constructive criticism, and have incorporated their feedback in our revision.
>
> **On the tradeoff between BAT accuracy on corrupted vs. clean data:**
> This is a common tradeoff in machine learning robustness, and has been previously explored in the adversarial setting [1]. Though our setting is different, the general argument of [111] still applies here: the optimal robust classifier may learn fundamentally different features than the optimal clean-data classifier (i.e. a “no free lunch” situation). With BAT, by introducing corruptions into the training data, we are not just optimizing for accuracy on clean videos, but on corrupted videos as well. Thus, our main goal in this work is to test a variety of baselines to understand how they perform in this new setting.
>
> **About model choice:**
> We have thought about this as well; we consider investigating the performance of BAT on other architectures as exciting future work. For action recognition, we used a ResNet-18 following the setup of [2]; for multi-object tracking, we used a DLANet-34 following the setup of [3]. To explore this problem further, we are currently working on preliminary experiments with BAT using Temporal Segmented Networks [4], another action recognition architecture.
>
> **Do existing methods in signal processing (w/o ML) work?**
> We thank R1 for bringing this to our attention. To this end, we ran a set of experiments using checksums as a method for detecting corrupted data. Checksums ultimately discard significant quantities of clean or near-clean videos, showing that this technique is unable to discriminate corrupted data. We have included these new results in Appendix D.2 (pg. 16).
>
> As for error correction, there are already error resilience mechanisms built into H.264 codec (i.e. macroblock/slice reordering, error concealment), which the videos we corrupt use. However, even with the resilience mechanisms in H.264, we can end up with severely distorted videos (see Fig. 4 and 5, pg. 6). We have added the relevant details of the H.264 codec in Section 2.1 of the current draft (pg. 2).
>
> *Clarification on Table 1:*
> The footnote clarifies how we calculate accuracy. Since network/file corruptions result in videos becoming unplayable, we do not count such examples as “incorrect,” because these examples cannot be loaded into the model. As a toy example, suppose we classify 900/1200 videos correctly. Perhaps after corruption, 200 videos become unplayable; as a result, we might get 760/1000 correct. We felt this was the fairest way to isolate the effect of these corruptions on model robustness.
>
> [1] Tsipras, Dimitris, et al. "Robustness may be at odds with accuracy." arXiv preprint arXiv:1805.12152 (2018).
>
> [2] Kataoka, Hirokatsu, et al. "Would Mega-scale Datasets Further Enhance Spatiotemporal 3D CNNs?." arXiv preprint arXiv:2004.04968 (2020).
>
> [3] Zhan, Yifu, et al. "A Simple Baseline for Multi-Object Tracking." arXiv preprint arXiv:2004.01888 (2020).
>
> [4] Wang, Limin, et al. "Temporal segment networks: Towards good practices for deep action recognition." European conference on computer vision. Springer, Cham, 2016.

---

### Official Review · AnonReviewer2 · 2020-10-29
**The topic this paper tackled may be important for the real-world application, and the method presented in the paper might be effective. However, the paper lacks experiments and evidence to properly support the authors’ claim. Also, the technical novelty is not significant. Therefore, I chose "rejection" as an evaluation of this paper.  If all the issues below are fully addressed, I may reconsider my assessment of this paper.**

**Rating:** 4
**Confidence:** 2

**Review:**

The authors explored the robustness of video machine learning models to bit-level corruption. They investigated previous methods such as Out-Of-Distribution (OOD) detection and adversarial training and found that they are not effective enough to defense against the bit-level corruption.  Accordingly, this paper proposed a new framework, Bit-corruption Augmented Training (BAT), which utilizes the knowledge about corruption by bit-level data augmentation at the training stage. Also, the authors argue that the proposed method outperforms the previous methods in handling the bit-level corrupted dataset.

While the proposed method seems simple and more effective than the previous studies, the authors do not currently provide a sufficient amount of evidence to support their claim.

Pros
-	The proposed defense technique is effective on the bit-level corruption of videos. Also, it is simple to apply in real-world deployment.
-	This is the first work that addresses the robustness against bit-level corruption of videos.

Cons
-	The technical novelty of this paper is not significant. They proposed a new framework for robustness to video bit-level corruption. However, data augmentation method is the main technical contribution this paper proposes. I think adopting existing technique to bit-level corruption problem is not significantly novel.
-	There is insufficient evidence that the proposed method is better than previous studies. I think additional experiments on OOD detection are needed. The authors implemented only one method as a baseline, ODIN [1]. Since there exist many other state-of-the-art methods than ODIN, such as Mahalanobis [2] and Outlier exposure [3], they should also be compared. In addition, there is no explanation of an input preprocessing method proposed in ODIN. The authors need to articulate why they omit the preprocessing. Also, additional experiments on UCF101 by using corruption-agnostic and corruption-aware defenses would make the paper more convincing.
-	Furthermore, in Section 4.3, the rationale behind the importance of detecting low-level corrupted samples by OOD is not elaborated. As the authors mentioned, largely corrupted videos are likely to be misclassified, while the accuracy for videos with low-level corruption does not decrease.
-	For clarity, I recommend the authors to correct the minor typos in the paper. The adversarial training in Section 4.3 seems to be worse compared to the no-defense baseline by 8.6 points (not 8.1 points) on clean data.

[1] Liang et al., "Enhancing the reliability of out-of-distribution image detection in neural networks.", ICLR'18

[2] Lee et al., "A simple unified framework for detecting out-of-distribution samples and adversarial attacks.", NIPS'18

[3] Hendrycks et al., "Deep anomaly detection with outlier exposure.", ICLR'19

----------------------------------
After rebuttal:

I appreciate the authors for thoughtful response and additional experimental results, which are helpful for further understanding of the manuscript. Especially, the additional experiment on the recent OOD detection method addresses my concern about the evidence that the previous OOD studies are not sufficient for defending the bit-level corruption.

Unfortunately, I am still not sure about the technical novelty of this paper. I agree that the paper proposed a new problem setting, but I do not think that the technical novelty is significant, given the proposed approach of just applying the data augmentation simply at a bit level, rather than at a pixel level.

Due to this concern, I want to keep my rating of "4. Ok but not good enough - rejection" as it is.

---

> ### Author Response · Authors · 2020-11-18
> **[draft updated] Clarifications on OOD, novelty. Thanks for the detailed comments and suggestions!**
>
> We appreciate R2’s comments on the **simplicity of our methods,** and for their recognition that the problem of robustness to bit-level network/file corruptions constitutes a **new problem setting.**
>
> **On novelty of the method:**
> We see ourselves as a first step in exploring robustness of video machine learning models to bit-level network/file corruptions. Thus, our primary novelty is in our **new problem setting of robustness to bit-level network/file corruptions.** Thus, we are evaluating the performance of multiple defenses as baselines. The best performing baseline is bit-corruption augmented training (BAT), a defense that assumes that we have prior knowledge of the severity of network/file corruptions seen during evaluation. Ultimately, we hope that this work is a starting point for future explorations.
>
> **On additional OOD experiments:**
> To address this, we run some additional experiments on energy-based outlier detection [1]. The results are almost the same as those from ODIN, in that many clean examples are ultimately discarded incorrectly. We include results on energy-based outlier detection in our new revision.
>
> Other out-of-distribution (OOD) detection techniques, like outlier exposure [2], requires access to out-of-distribution data; in contrast, the OOD detection methods we evaluate (score-based OOD detection) make no assumptions about access to OOD data and only use the model output. A method inspired by [2] would be an interesting direction for future work in corruption-aware OOD detection; however, our inclusion of OOD methods is intended to reflect a situation in which one only has access to the model logits.
>
> **Why do we run OOD detection at low levels of corruption?**
> The point of running OOD detection experiments is to demonstrate that the OOD methods tested mistakenly mark many in-distribution examples as out-of-distribution, meaning that OOD detection discards many clean examples from the data. This is a key limitation of ODIN and energy-based OOD in this setting, and suggests that these methods are not a suitable overarching defense against all types of corruptions evaluation.
>
> **What is the input pre-processing method for OOD detection:**
> We follow the evaluation setup described in Appendix F.2 (pg. 19). As a brief summary; the clip is segmented into contiguous 16-frame segments with the ActivityNet mean subtracted, then each segment is passed through the model following the setup of [3]. The ODIN procedure then proceeds as described in Section 3.1 (pg. 3-4).
>
> Lastly, we thank R2’s diligence in catching our typo; we have fixed that error.
>
> [1] Liu, Weitang, et al. "Energy-based Out-of-distribution Detection." Advances in Neural Information Processing Systems 33 (2020).
>
> [2] Hendrycks, Dan, Mantas Mazeika, and Thomas Dietterich. "Deep anomaly detection with outlier exposure." arXiv preprint arXiv:1812.04606 (2018).
>
> [3] Kataoka, Hirokatsu, et al. "Would Mega-scale Datasets Further Enhance Spatiotemporal 3D CNNs?." arXiv preprint arXiv:2004.04968 (2020).

---

### Author Response · Authors · 2020-11-18
**Summary of revision**

We thank the reviewers for their comments. We have responded to each reviewer in detail as well. We are encouraged that they find this problem space to be interesting (R4) and underexplored (R1, R3). We are also glad to see positive feedback on the effectiveness (R2, R3) and simplicity (R1, R2, R3) of our method. Furthermore, we are glad that R4 appreciated the realisticness of the corruptions we simulate. We have incorporated their feedback into our draft. We recap our main goal and summarize changes made below.

**Recap — Significance of the work.** In this paper, we take a first step in exploring the robustness of video machine learning models to bit-level network/file corruptions. Our primary novelty is in the problem setting of robustness to bit-level network/file corruptions, which **almost all reviewers agreed was underexplored and new.** To this end, we evaluate the performance of multiple baseline defenses in this new setting.

In addition, to the best of our knowledge, **BAT is the first to explore augmentation in the bit space,** whilst many previous works have primarily focused on pixel space augmentation. This is both non-trivial and new, as the augmentation undergoes additional encoding and decoding processes. Ultimately, we see our work as a starting point for future explorations of this problem space.


**Summary of changes**

**[R2, R3, R4 → ]** Re: concerns about technical novelty, we have emphasized in our individual responses that our primary novelty is in the setting of bit-level network/file corruption, which most reviewers agreed was new/underexplored. Thus, this work is meant to be a starting point for future explorations of this problem setting. We have revised the writing to make our intended framing more clear.

**[R1, R4 → ]** We have run additional results on system-level fault tolerance methods like checksums for filtering corrupted videos, reporting results in the Appendix D.2 (pg. 16). Checksum-based detection is ultimately ineffective, since this results in many clean/near-clean videos being removed mistakenly. We also added an explanation of the inbuilt error resilience features of H.264, the video codec that we use, in Section 2.1 (pg. 2).

**[R2 → ]** We added additionally the latest out-of-distribution detection technique in literature, and report the results in the Appendix.

**[R1 → ]** We briefly address the tradeoff between BAT accuracy and accuracy on clean data in Section 4.4 (pg. 7-8).

**[R3, R4 → ]** We have emphasized more strongly throughout the paper the real-world inspiration for the corruptions we simulate, and highlighted the assumptions that each baseline defense makes. We are also in the process of studying corruptions resulting from network failure modes calibrated to real-world network links, such as those simulated in Pantheon [1].

In summary, we believe that our work contributes to the field of machine learning robustness by studying model robustness in an underexplored setting of bit-level network/file corruptions. We evaluate the performance of reasonable baselines in this setting, and hope that our work opens up promising new directions for future studies of model robustness to bit-level corruptions.

We would like to thank all the reviewers again for their constructive comments and feedback, which helped us improve our draft.

[1] Yan, Francis Y., et al. "Pantheon: the training ground for Internet congestion-control research." 2018 {USENIX} Annual Technical Conference (USENIX ATC 18). 2018.

---

### Decision · Program_Chairs · 2021-01-07
**Final Decision**

**Decision:**

Reject

**Comment:**

This paper investigates robustness of the neural networks under bit-level network and file corruptions, and proposes corruption-agnostic and corruption-aware defense approaches. The Bit-corruption Augmented Training is introduced, which is about applying the data augmentation at a bit level.

The majority of the reviewers are against the acceptance of the paper. R1 gave the rating of "marginally above acceptance threshold", finding the problem interesting but not the proposed solution. The other reviewers gave rejections and a clear reject rating.

The main concern raised by all of the reviewers is regarding the technical novelty of the proposed approach in the paper. While some reviewers appreciate the importance of the problem and the thoroughness of the experiments more than the others, none of the reviewers find the proposed solution novel and interesting.

The AC agrees with the reviewers that the technical contribution of the paper is not significant despite that it focuses on an interesting problem. We do not recommend this paper to ICLR.